# Functional and Allergenic Properties Assessment of Conalbumin (Ovotransferrin) after Oxidation

**DOI:** 10.3390/foods11152308

**Published:** 2022-08-02

**Authors:** Liangtao Lv, Liying Ye, Xiao Lin, Liuying Li, Jiamin Chen, Wenqi Yue, Xuli Wu

**Affiliations:** Health Science Center, School of Public Health, Shenzhen University, Shenzhen 518060, China; hdsgllt@163.com (L.L.); 2060243014@email.szu.edu.cn (L.Y.); 2070245003@email.szu.edu.cn (X.L.); 2110245009@email.szu.edu.cn (L.L.); 2100243004@email.szu.edu.cn (J.C.); 2200243010@email.szu.edu.cn (W.Y.)

**Keywords:** conalbumin, oxidation, function, digestibility, allergenicity

## Abstract

Conalbumin (CA) is an iron-binding egg protein that has various bioactivities and causes major allergenicity in humans. This study investigated how oxidation affects the multiple functional properties of CA. The lipid peroxidation method was used to prepare treated CA [2,2′-azobis (2-amidinopropane) dihydrochloride (AAPH)-CA and acrolein-CA] complexes. CA induced structural changes through oxidation. These changes enhanced the digestibility, rate of endocytosis in dendritic cells, and emulsifying and foaming properties of CA. ELISA and immunoblot analysis showed that the complexes reduced the IgE-binding ability of CA through lipid oxidation. KU812 cell assays showed that modification by AAPH and acrolein caused the release of IL-4 and histamine to decline. In conclusion, oxidation treatment modified the functional and structural properties of CA, reducing allergenicity during processing and preservation.

## 1. Introduction

Chicken eggs are a popular food globally because they are easily available and nutritious. Egg proteins have several important functional properties, including emulsification, foaming, and allergenicity [1,2]. The functional properties of egg protein are influenced by different food-processing methods, including heat treatment [3], microwaving [4], high-pressure [5], and protein glycation [6]. Eggs are also major allergy-causing food items in children [7,8], with a variety of allergic symptoms including hives, rashes, and atopic dermatitis.

A complex combination of factors influences the allergic responses of people to food containing eggs. Consequently, it is not sufficient to, simply, alter the functional properties in egg protein. Chemical reactions are major factors that influence food allergenicity during food processing. Thus, ideally, the functional properties of egg products should be evaluated from the perspective of food oxidation, which potentially arises during processing. Eggs are highly prone to oxidation by polyunsaturated fatty acids (PCA) during food processing [9]. PCA peroxidation is a reaction that is catalyzed by free radical-mediated complex chains. For instance, soy protein is easily modified by the generation of peroxyl in solutions containing 2,2′-azobis (2-amidinopropane) dihydrochloride (AAPH) under aerobic conditions [10]. Moreover, acrolein is as an important, abundant aldehyde that is created from the main secondary products of peroxidation [11]. Oxidation also binds and modifies the structure of proteins [12], altering the physical and chemical properties of processed seafoods [13]. Protein oxidation strongly affects food quality. It influences the nutrition and functional properties of proteins [14]. Moreover, protein oxidation alters the structure and allergenicity of food [15,16].

Conalbumin (CA) (also called ovotransferrin) is a monomeric glycoprotein that makes up about 12–13% of egg white. It has a molecular weight of 76 kDa, with 15 disulfide bonds and 686 amino acids [17]. Egg white has several important functional properties, including foaming and emulsification. However, CA is also main allergic proteins of egg white and yolk [18]. Therefore, oxidative modifications to egg whites might affect the functional properties of CA. To date, knowledge remains limited of how lipid peroxidants impact the allergenicity and functional properties of CA.

This study investigated whether the lipid peroxidation of egg white alters its functioning, digestion, and allergenicity. Structural changes were analyzed by circular dichroism (CD), intrinsic fluorescence (IF), and differential scanning calorimetry (DSC). Functional properties were detected by examining emulsifying and foaming properties. IgE-binding activity was analyzed by enzyme-linked immunosorbent assay (ELISA) and Western blot using egg-white-allergic patients’ sera. Simulated gastric digestion was used to analyze the digestive stability of AAPH-CA and acrolein-CA. The dendritic cell (DC) model and KU812 cells model were used to evaluate the rate of endocytosis and allergenicity of modified protein, respectively. The results are expected to provide new insights on how lipid oxidation during food processing alters the functional and allergenic properties of eggs.

## 2. Materials and Methods

### 2.1. Materials

Conalbumin (CA), AAPH, and acrolein were purchased from Sigma Chemical Co. (St. Louis, MO, USA). ELISA kits of histamine and IL-4 was acquired from R&D systems (Minneapolis, MN, USA). pHrodo Green was purchased from Invitrogen (Breda, The Netherlands). All other reagents were of analytical grade unless stated otherwise.

### 2.2. Human Sera 

Sera samples from eight patients who were allergic to eggs were provided by Shenzhen Children’s Hospital, China. This study was approved by the Human Ethics Committee of Shenzhen University (202009026). Sera IgE antibodies to egg were determined by the immunoCAP system (phadia AB, Uppsala, Sweden). Serum was collected from allergic patients (≥10 kU/L) and was then stored at −40 °C until use.

### 2.3. Preparation of Oxidation Samples

Purified CA was dissolved in diluted in phosphate buffer solution (PBS, pH 7.4). The concentration of proteins was adjusted to 5 mg mL^−1^. Then, the CA solution was mixed in AAPH and acrolein solution (*v*/*v* = 1:1) for 24 h at 37 °C. The final concentrations of AAPH and acrolein were set at 1 mmol L^−1^, 5 mmol L^−1^, 25 mmol L^−1^ and 0.1 mmol L^−1^, 1 mmol L^−1^, and 10 mmol L^−1^, respectively. Unreacted AAPH and acrolein were removed by dialysis for 48 h at 4 °C. After the reaction, the total solution was collected and lyophilized. The final samples were stored at −40 °C until use.

### 2.4. Electrophoretic Analysis

SDS-PAGE was carried out according to a method described by Laemmli et al. [19]. The treated CA solution was heated for 7 min before loading it on 5% stacking gel and 12% separating gel. After electrophoresis, the gels were stained with Coomassie Brilliant Blue R-250G overnight and were subsequently destained using deionized water. The results of the gel were acquired using a ChemiDoc XRS Imaging System (Hercules, CA, USA).

### 2.5. Structure Analysis

#### 2.5.1. Intrinsic Fluorescence Measurements

Intrinsic fluorescence measurements were carried out according to the method described by Lv et al. [20]. The samples were diluted to 0.5 mg mL^−1^ in PBS. Then, all the samples were detected by a RF-5301PC fluorescence spectrophotometer (Kyoto, Japan). The parameters were set at: 280 nm excitation wavelength; 290–400 nm emission wavelengths; 2.5 nm slit width; and 8 nm s^−1^ scanned speed.

#### 2.5.2. Secondary Structure Analysis

The secondary structure of different treated samples after oxidation was detected by a JASCO J-815 CD spectropolarimeter (Tokyo, Japan). The sample solution was diluted to 0.5 mg mL^−1^. The parameters were set at: 190 to 260 nm spectral range; 1.0 nm bandwidth; 100 nm/min scan rate; and 0.25 s interval. 

### 2.6. Thermal Stability Analysis

DSC was used to determine the thermal stability of CA. CA was treated using DSC-60 apparatus (Shimadzu, Tokyo). The parameters were set at: 40 to 120 °C and 5 °C min^−1^ scan rate.

### 2.7. Functional Properties Analysis

#### 2.7.1. Emulsifying Properties

Emulsifying properties were performed using previously reported methods [21], with some modifications. In brief, 10 mL soybean oil was added to 30 mL sample solution (2 mg mL^−1^). The final solution was then homogenized for 2 min. Subsequently, 50 μL of two homogenized samples was immediately diluted with 0.1% SDS solution at 0 and 10 min, respectively. Absorbance was then recorded at 500 nm. The emulsion active index (EAI) and emulsion stability index (ESI) were calculated as follows:EAI (m^2^/g) = (2 × 2.303 × A_0_ × DF)/(c × φ × 10,000)
ESI (%) = (A_0_/Δ_10_) ×100

The parameters were set at: A_0_ and A_10_ represent absorbance at 0 and 10 min, respectively, after homogenization; C represents protein concentration (g/mL); DF represents the dilution factor; and φ represents the oil volume fraction (*v*/*v*) of the emulsion (φ = 0.25).

#### 2.7.2. Foaming properties

Foaming properties were determined following existing methods [22,23]. All samples (2 mg mL^−1^) were stirred at 3000 rpm for 10 min. Then, the total volume of each sample was detected at 0 and 30 min, respectively. Foam stability (FS) and foaming capacity (FC) were calculated as follows:FS (%) = (volume after standing − volume before stirring)/volume before stirring
FC (%) = (volume after stirring − volume before stirring)/volume before stirring

### 2.8. Assessment of In Vitro Allergenicity on CA Oxidation Complexes

#### 2.8.1. Western Blot

The treated samples were electrophoretically transferred to polyvinylidene fluoride (PVDF) membranes after SDS-PAGE. First, the membranes were blocked using 1% BSA in PBST (0.05% Tween 20 in PBS) for 90 min. After washing with PBST three times, the membranes were incubated with pooled human sera (*v*/*v* = 1:40) in PBST. After washing again, the membranes were incubated with goat anti-human IgE antibody (Kirkegaard & Perry Laboratories, Gaithersburg, MD, USA) (diluted 1:5000) in PBST. Finally, the membrane was washed three times again. The membrane was then incubated with ECL Reagent (GE Healthcare, Buckinghamshire, UK) and exposed to X-ray film to obtain the results.

#### 2.8.2. ELISA Analysis

The solutions of CA samples subjected to different treatments (100 μL) were incubated with 50 mmol/L sodium carbonate buffer (pH 9.6) in a 96-well plate at 4 °C overnight. After washing the samples three times with PBST, they were blocked with 1% BSA in PBST for 90 min at 37 °C. After washing the samples three times with PBST, they were incubated with 100 μL pooled human sera (diluted 1:50) at 37 °C for 90 min. After washing again, the samples were incubated with 100 μL horseradish peroxidase enzyme (HRP)-labeled goat antihuman IgE antibody (diluted 1:10,000) for 60 min at 37 °C again. After a final wash with PBST, 100 μL TMB was added to the plate, and it was incubated at 37 °C for 20 min. Finally, the reaction was terminated by 2 mol/L sulfuric acid. Finally, absorbance was measured at 450 nm. 

#### 2.8.3. Assays of KU812 Cells

According to a previous method [24], KU812 cells were cultured in complete RPMI 1640 medium supplemented with 100 μg mL^−1^ streptomycin, 100 U mL^−1^ penicillin, 2 mmol L^−1^ L-glutamine (90%) and FBS (10%). KU812 cells (1 × 10^7^ cells/mL) were incubated with egg white and the serum of allergic patients (100 μL) for 24 h. It was then incubated with 10 μL of CA samples from different treatments (5 mg/mL) for 4 h. Finally, the cell supernatant was used to detect IL-4 and histamine content with ELISA kits after centrifugation.

### 2.9. Assay of Dendritic Cells (DCs)

Bone marrow cells from mice were cultured in complete RPMI 1640 supplemented with 5 ng/mL of IL-4 and 10 ng/mL of granulocyte-macrophage colony-stimulating factor. After culturing the cells for 6 days, more than 90% expressed high levels of major histocompatibility complex class II (MHC-II) and CD11c. CA, AAPH-CA, and acrolein-CA labeled by pHrodo Green were incubated with cells (10^6^ cells/mL) to assess protein endocytosis. At 0, 15, and 30 min, a microplate reader was used to detect pHrodo-Green-positive cells.

### 2.10. In Vitro Digestion Stability Assay

The digestibility assay of different CA samples was determined based on existing methods with minor modification [25]. The CA and treated CA (0.5 mg/mL) were added to simulated gastric fluids, and digestion was performed for different time intervals (0, 1, 5, 10, 20, 30, 60, 90, and 120 min) at 37 °C. The final samples were assessed to determine digestibility and IgE-binding activity using SDS-PAGE and Western blot analysis (using goat anti-human IgE antibodies), respectively.

### 2.11. Statistical Analysis

The obtained results were analyzed by ANOVA using Duncan’s test with Prism 7.0. Values that differed significantly (*p* ≤ 0.05) were considered. At least three replicates of all measurements were evaluated.

## 3. Results 

### 3.1. SDS-PAGE

CA oxidation was determined by SDS-PAGE (Figure 1). When AAPH concentrations were 5 and 25 mmol/L, the treated CA bands migrated upwards (Figure 1, lane 2, 3). When acrolein concentrations were 1 and 10 mmol/L, higher molecular weight proteins were detected (of ~152 kDa, 228 kDa, and above; Figure 1, lanes 5, 6). CA bands aggregated more with increasing acrolein concentration, whereas band intensity declined to 76 kDa (Figure 1). The bands did not change when CA was modified with AAPH concentration of <1 mmol/L and acrolein concentrations of <0.1 mmol/L (Figure 1, lanes 1, 4).

### 3.2. Effect of Oxidation on CA Structure

CD spectra showed that CA samples occurred at a ~212 nm negative peak, which represented α-helical protein structures (Figure 2A). Compared to untreated CA, the absolute θ values of treated CA clearly declined on this negative band. Thus, the α-helical content of CA decreased after oxidation, which might cause the protein structure to unfold.

Excitation of intrinsic fluorescence at 280 nm (Figure 2B) generated a maximum emission spectrum of 312 nm. Compared to untreated CA, the intrinsic fluorescence intensity of oxidation complexes significantly declined with increasing AAPH and acrolein concentrations. The intrinsic fluorescence intensity of treated CA reached approximately 2.8% and 42.3% that of the untreated protein when CA was modified with 25 mmol/L AAPH and 10 mmol/L acrolein, respectively. The maximum emission wavelength also showed a blue shift (from 336 to 315 nm and from 336 to 332 nm, respectively).

DSC thermograms of untreated CA had one endothermic peak at 92.5 ± 0.5 °C. After oxidation with different concentrations of AAPH, the thermal denaturation temperatures of CA were 62.7 ± 0.5, 56.0 ± 0.3, and 66.9 ± 0.3 °C. After oxidation with different concentrations of acrolein, the thermal denaturation temperatures of CA were 68.7 ± 0.3, 58.1 ± 0.5, and 56.4 ± 0.5 °C (Figure 2C). Thus, the thermal stability of CA significantly decreased after oxidation.

### 3.3. Function Properties

Emulsifying activity was greater for treated CA compared to untreated CA (Figure 3A). Compared with untreated CA, acrolein-CA and AAPH-CA had higher emulsifying activity. After oxidation with AAPH and acrolein, the conformational structure of CA might alter with exposed hydrophobic domains, which might influence the emulsifying activity of proteins. However, emulsifying stability declined (Figure 3B). Comparison of emulsifying properties showed that both the FC and FS of the oxidation samples were significantly (*p* < 0.05) higher compared to those of purified OVA (Figure 3C).

### 3.4. Stability of CA after Oxidation in a Simulated Digestion In Vitro

SDS-PAGE showed that, within 1 min, the bands of unmodified CA were degraded by pepsin into several small bands (approximately 38, 30, 15, and 10 kDa) (Figure 4). The CA band was almost completely degraded after 5 min (Figure 4A). With increasing digestion time, only some small bands (approximately 15 and 10 kDa) remained after 120 min. The 5 mmol/L AAPH-CA band was almost completely degraded after 1 min incubation with pepsin (Figure 4B). After 60 min, the acrolein-treated CA band was similar to that of untreated CA, whereas the aggregation band of CA was almost completely degraded (Figure 4C). Thus, the digestibility of CA increased after AAPH and acrolein treatment. SDS-PAGE produced limited information on the simulated gastric digestion of CA and oxidation treatment of CA. The results for the Western blotting of gastric digestion were similar to those of SDS-PAGE (Figure 4D–F). 

### 3.5. Uptake by DCs

pHrodo Green was used to label CA, AAPH-CA, and acrolein-CA to identify where they were located in the endolysosomal compartments. The rate of endocytosis of native protein was slower (Figure 5). The rate of endocytosis at 30 min showed no significant improvement (Figure 5). The rate of endocytosis of modified protein increased with increasing AAPH and acrolein concentration.

### 3.6. In Vitro Allergenicity Assessment

Western blot showed that the CA band weakened with increasing AAPH and acrolein concentration (Figure 6A); thus, IgE-binding abilities of CA appeared to decline after oxidation. CA aggregations were still detected after acrolein treatment. ELISA showed that the IgE-binding capacity of CA clearly declined after AAPH and acrolein treatment (Figure 6B). The IgE-binding ability of CA supported the SDS-PAGE results. Compared to untreated protein, treated CA had lower histamine (Figure 6C) and IL-4 (Figure 6D) levels after oxidation; thus, the ability of CA to trigger cell degranulation declined after oxidation. 

## 4. Discussion

Lipid peroxidation potentially alters the functional properties of egg whites, including allergenicity [26]. This study used SDS-PAGE to monitor structural changes of CA after oxidation. The results showed that treated CA bands formed aggregations after oxidation. This might be attributed to β-mercaptoethanol breaking the protein disulfide bonds in the sample buffer before SDS-PAGE. Furthermore, the lipid peroxidation products and amino acid residues of CA might be linked by non-disulfide covalent linkages. Thus, CA oxidation could induce modifications to CA structure, which might impact its allergenicity. The SDS-PAGE results of treated CA with lipid peroxidation products supported those obtained by previous research [27].

Far-ultraviolet CD spectra were performed to evaluate how oxidation influenced the secondary structure of the protein, which includes β-sheet, α-helix, β-turn, and a random structure [28]. Treated CA aggregations formed more complex structures after AAPH and acrolein treatment. Oxidation might cause the intensity of the peaks to decline, indicating that the proportion of the α-helix in CA declined. The results supported those of previous studies [16,20]. Van der Waals interactions are disrupted when amino acid residues were treated by lipid peroxidation products, which might explain our results. 

Intrinsic fluorescence spectra were used to investigate the conformational structure of the protein. Lipid peroxidation might alter the fluorescence intensity of allergens. The tyrosine, tryptophan, and phenylalanine residues of proteins contribute to the intrinsic fluorescence spectra [16,20,29]. The intensity of the fluorescence decreased due to changes in the microenvironment, such as tyrosine residues possibly being disrupted after oxidation or amino acids being less exposed to the aqueous environment because of protein coagulation [30].

The thermal stability of CA significantly decreased after oxidation, possibly because oxidation denatured CA. Protein oxidation is also accompanied by the destruction of the conformational structure [31,32], affecting both physicochemical and functional properties [32]. Narayan et al. [33] reported the loss of the tertiary structure, reduction of disulfide bonds, and increased accessibility of reactive glutamine and lysine residues in the protein substrate, all of which altered the thermal stability of the protein.

The functional properties of egg white might also be influenced by oxidation with AAPH and acrolein. A stable emulsion has substantial potential as a delivery system in foods [34]. Evaluation of functional properties showed that oxidation could increase the random coiling of CA, enhancing the emulsifying activity of the protein [35]. However, increased random coiling could also cause emulsifying stability to decline. Foaming capacity (FC) might be associated with the creation of air droplets in the protein, whereas foaming stability (FS) ensures that the protein contains adequate viscosity to sustain FC and stop coalescence or cracking. The conformation of proteins is the main factor that affects their emulsifying properties [21]. Because the tertiary conformation stability of CA decreased, surface hydrophobicity and molecular flexibility were enhanced, thus improving the characteristics of foam forming and stabilizing. 

The digestibility of modified CA was assessed by simulated gastric fluids. According to the length of time that ingested food typically remains in the stomach, the digestion duration for simulated gastric fluids was set to 2 h. The results showed that the peptic digestive patterns of CA differed following oxidation (Figure 4). Pepsin cleaves to Phe or Tyr residues to form peptide bonds [36]. CA was degraded; however, simulated gastric digestion caused a clear decrease in oxidized CA. This phenomenon might be attributed to the accessible digestion sites of CA being altered by protein oxidation. Oxidation might expose more digestion sites on the protein and subsequently modify the allergenic epitopes on the CA surface, thus accelerating pepsin digestion. Treatment with proteases is effective at reducing the IgE-binding capacity of CA. Smaller CA proteolytic fragments (approximately 15, 10 kDa) were generated by simulated gastric digestion bound with IgE, possibly because the immunogenic IgE-binding epitopes in these fragments were retained following the digestion process. Thus, the digestibility of CA likely increased after oxidation because oxidation caused CA to unfold. After digestion, the IgE-binding ability of oxidized CA was lower compared to that of digested unmodified CA. Furthermore, oxidation caused the digestibility of CA to increase, which could explain the endocytosis of pHrodo-Green-labeled AAPH-CA, acrolein-CA, and untreated CA. pHrodo Green fluoresces brightly at acidic pH but does not fluoresce at neutral pH. Compared to untreated CA, AAPH-CA and acrolein-CA were easier to digest. Thus, AAPH-CA and acrolein-CA could be processed faster in DCs. At acidic pH in the lysosome of DCs, pHrodo Green fluoresced brightly, resulting in a stronger fluorescence intensity of the modified protein.

The present study showed that the IgE-binding capacity of CA was noticeably reduced in the 25 mmol/L AAPH and 10 mmol/L acrolein treatment, possibly due to structural changes caused by oxidation. Previous studies showed that the amino acid residues of the protein (Lys, Try, and His) are easily modified by lipid peroxidation [37]. Peptides 1–24, 40–60, 142–159, and 139–117 are major IgE-binding epitopes of CA [38], which contains many Lys, Try, and Leu residues. Therefore, lipid peroxidation led to the destruction of allergic epitopes, which could reduce the IgE-binding capacity of CA. 

In addition, a KU812 cells model was used to evaluate the potential allergenicity of AAPH-CA, acrolein-CA, and untreated CA. IgE-mediated KU812 cells are frequently used to examine type I allergic responses [39]. These assays are commonly used to quantify variation in IgE-binding capacity. The magnitude of allergic symptoms depends on the secretion of cellular components. Consequently, the assay provides information on body condition in an allergic state [40,41]. The present study showed that oxidation-modified CA noticeably inhibited the release of IL-4 and histamine. In particular, oxidation might inhibit the release of cytokines and mediators. A previous study showed that the degranulation capacity of KU812 cells declined when bovine α-lactalbumin was exposed to gamma irradiation [42]. 

The modification of egg CA via oxidation altered the conformational structure of CA, potentially altering antigenic epitopes through intra/intermolecular interactions. Previously, we found that modifying the conformational structure of TM via oxidation inhibited IgE-binding activity [26]. The lower degranulation capacity of KU812 cells could be elucidated by oxidation covering/destroying IgE epitopes and changes to other concomitant structural properties, which hindered the degranulation of mast cells and basophils, suppressing allergic responses. Park et al. [43] reported that ovalbumin lost allergenicity following N-acetylglucosaminidase treatment. This phenomenon might be explained by chemical changes covering or destroying the IgE epitope and altering the immunological features, which reduced its allergen-inducing capacity. These results provide new insights on how allergenicity changes during food preservation and processing.

## 5. Conclusions

CA is a major egg protein that is widely used in the food industry. However, oxidation changes the structural and functional properties of CA. Compared to untreated CA, oxidation lowers the IgE-binding capacity of CA, releasing histamine and IL-4 from KU812 cells. This phenomenon potentially reduces the allergenicity of CA. In conclusion, the current study demonstrated that oxidation treatment modifies the functional and structural properties of CA, reducing allergenicity during processing and preservation.

## Figures and Tables

**Figure 1 foods-11-02308-f001:**
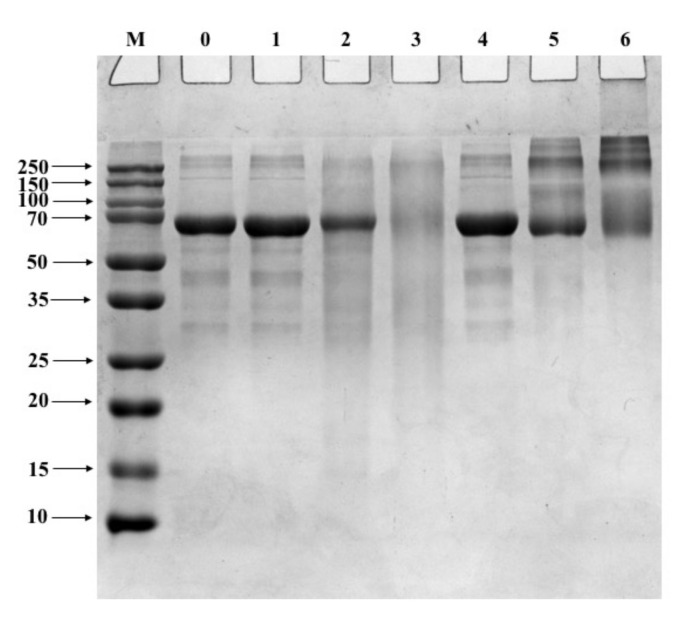
SDS-PAGE analysis of AAPH-CA and acrolein-CA. An equal amount of CA (5 µg) was loaded to each lane. The molecular mass of the standard protein marker ranged from 10 to 250 kDa. Lane M, protein marker; Lane 0, native CA; Lanes 1–3, AAPH concentrations 1, 5, and 25 mmol/L, respectively; Lanes 4–6, acrolein concentrations 0.1, 1, and 10 mmol/L, respectively.

**Figure 2 foods-11-02308-f002:**
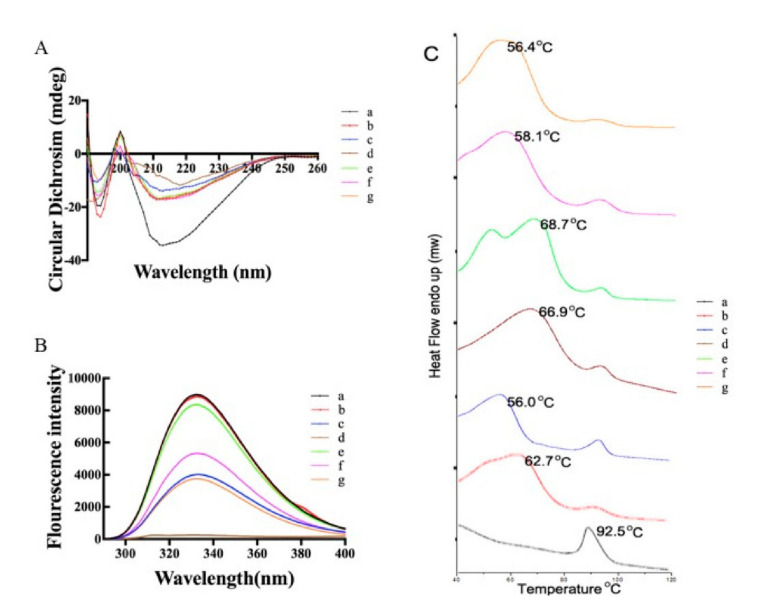
Effect of AAPH-CA and acrolein-CA on: (**A**) CD spectroscopy, (**B**) intrinsic fluorescence absorption, and (**C**) DSC thermograms. Figure (**C**) a–g represents: (**a**) control (native CA); (**b**) 1 mmol/L AAPH; (**c**) 5 mmol/L AAPH; (**d**) 25 mmol/L AAPH; (**e**) 0.1 mmol/L acrolein; (**f**) 1 mmol/L acrolein; and (**g**) 10 mmol/L acrolein.

**Figure 3 foods-11-02308-f003:**
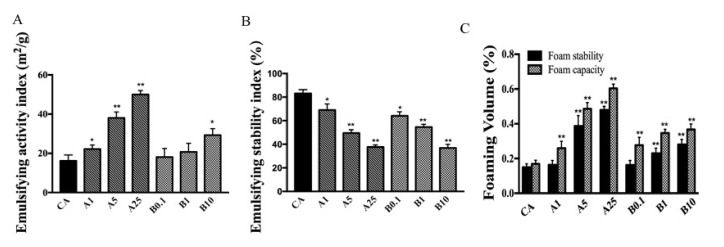
(**A**) Emulsifying activity, (**B**) emulsifying stability, and (**C**) foaming properties of CA, AAPH-CA, and acrolein-CA. * represents significant differences when compared with CA; * *p* < 0.05; ** *p* < 0.01.

**Figure 4 foods-11-02308-f004:**
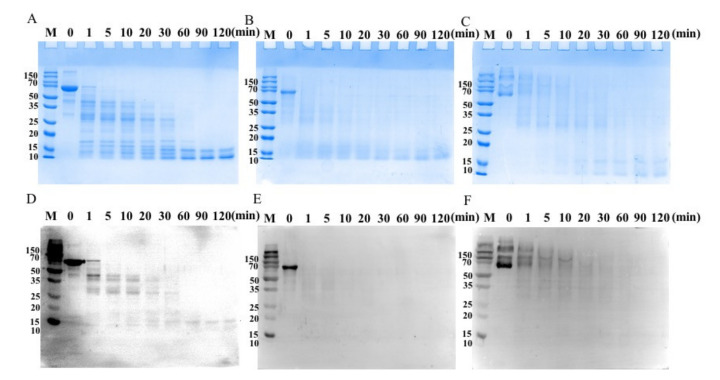
Effect of simulated gastric fluid digestion on the purified CA egg treated with 5 mmol/L AAPH and 1 mmol/L acrolein. Enzymatic digestion was performed, followed by SDS–PAGE. (**A**) Egg CA by pepsin; (**B**) AAPH-CA by pepsin; (**C**) acrolein-CA by pepsin. Enzymatic digestion was performed, followed by western blot analyses. (**D**) CA, (**E**) AAPH-CA, and (**F**) acrolein-CA were detected using pooled sera from patients allergic to egg with pepsin. In the controls (con), protease was replaced with 25 mmol/L PBS (pH 7.5). Molecular mass of protein markers (M) is shown (left side of each slide).

**Figure 5 foods-11-02308-f005:**
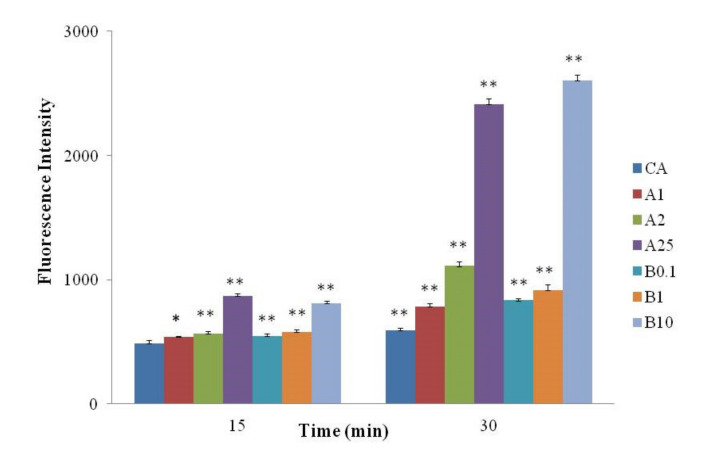
Uptake of pHrodo-Green-labeled CA, AAPH-CA, and acrolein-CA by DCs. * represents significant differences when compared with CA, * *p* < 0.05; ** *p* < 0.01.

**Figure 6 foods-11-02308-f006:**
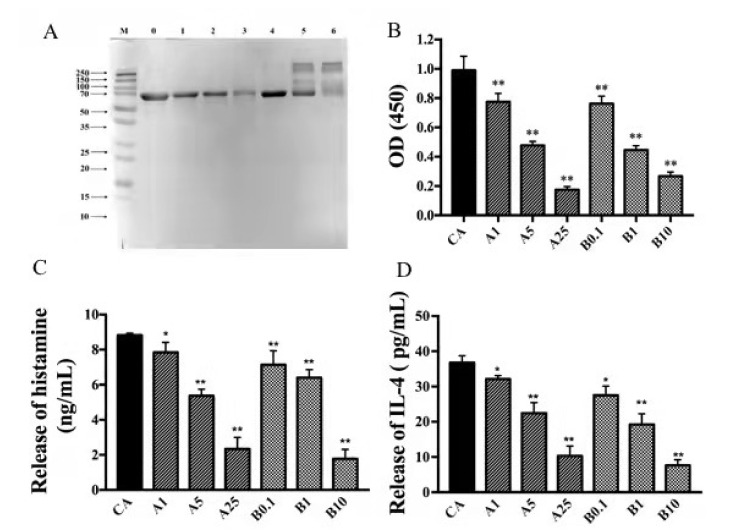
In vitro allergenicity assessment of CA oxidation complexes. (**A**) IgE-binding capacity of AAPH-CA and acrolein-CA in immunoblot analysis using pooled sera (IgE) from patients allergic to egg. (**B**) Indirect ELISA to detect the IgE-binding capacity of AAPH-CA and acrolein-CA using human serum. (**C**) Basophil histamine release with KU812 cell degranulation and (**D**) IL-4 release with KU812 cell degranulation using CA, AAPH-CA, and acrolein-CA. * *p* < 0.05; ** *p* < 0.01.

## Data Availability

No new data were created or analyzed in this study. Data sharing is not applicable to this article.

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
