# Peer review of "Functional and Allergenic Properties Assessment of Conalbumin (Ovotransferrin) after Oxidation"

_foods, 2022, doi:10.3390/foods11152308_

Round 1

Reviewer 1 Report

The subject of the Manuscript entitled “Functional and allergenic properties assessment of conalbumin (ovotransferrin) after oxidation” by Liangtao Lv, Liying Ye, Xiao Lin, Liuying Li, Jiamin Chen, Wenqi Yue, and Xuli Wu is relevant and can be considered for publication in MDPI Foods after the major revision of the manuscript. The authors put effort and performed relevant experimental work to explain changes in protein structure, digestibility, allergenicity, and functional properties of conalbumin upon oxidation. In this manner and order, I suggest reorganization and better presentation of the obtained results.

I suggest some points be revised or explained:

1.     The manuscript requires detailed technical editing in terms of typographical error correction (example: page 2, line 45 “allergencity” should be corrected to allergenicity, etc.). Also, editing of the English language should be done (example: page 5, line 166 “was used to detected” should be corrected to was used to detect, etc.).

2.     In materials and method 2.8.2. ELISA analysis, authors used goat antihuman IgE antibody as the secondary antibody, is this Ab conjugated with horseradish peroxidase enzyme (HRP) since TMB substrate was used?

3.     For in vitro allergenicity assessment KU812 cell line was used which is well known for its ability to differentiate into basophils or eosinophils as a response to different stimuli. Do the authors include some tests to confirm that KU812 used in the experiment exhibited the morphology of basophils? Such as expression of surface marker antigens CD13 and CD15 (CD13+ /CD15 cell population is designated as basophils, CD13+ /CD15+ cell population is designated as eosinophils)?

      4.    To assess protein endocytosis the authors used mouse bone marrow-derived DCs, in this experiment do the authors include the step of washing cells upon incubation with fluorescently labeled proteins? Residual fluorescently labeled proteins could interfere with obtained results. Also Figure 5., in which these results are presented, I suggest being modified in a way that fluorescence intensity should be presented as bars and figure legend for different protein treatments should be added.

Author Response

Dear reviewer:

Thank you so much. We have carefully revised and checked our manuscript and answered all questions point by point. We highlighted the changes using the “Track Changes” function. We greatly appreciate your thoughtful advice and comments to help improve our study. We would like to re-submit the manuscript for consideration of publication. We would assure that all of the authors have read and approved the final submitted manuscript.

Q1.     The manuscript requires detailed technical editing in terms of typographical error correction (example: page 2, line 45 “allergencity” should be corrected to allergenicity, etc.). Also, editing of the English language should be done (example: page 5, line 166 “was used to detected” should be corrected to was used to detect, etc.).

Answer:Thank you for your detailed comments. page 2, line 45 “allergencity” was corrected to allergenicity. page 5, line 166 “was used to detected” was corrected to “was used to detect”. In addition, the editing of the English language had been modified carefully. We asked Dr. Ann Power Smith for editorial revision of this manuscript. Dr. Ann Power Smith is a professional English language editor for scientific and medical manuscripts and has 20 years' experience editing biomedical manuscripts, grants and other communications

Q2.     In materials and method 2.8.2. ELISA analysis, authors used goat antihuman IgE antibody as the secondary antibody, is this Ab conjugated with horseradish peroxidase enzyme (HRP) since TMB substrate was used?

Answer: Thank you for your good comments. page 4, line 148 “goat antihuman IgE antibody” should be “horseradish peroxidase enzyme (HRP)-labeled goat antihuman IgE antibody”.

Q3.     For in vitro allergenicity assessment KU812 cell line was used which is well known for its ability to differentiate into basophils or eosinophils as a response to different stimuli. Do the authors include some tests to confirm that KU812 used in the experiment exhibited the morphology of basophils? Such as expression of surface marker antigens CD13 and CD15 (CD13+ /CD15– cell population is designated as basophils, CD13+ /CD15+ cell population is designated as eosinophils)?

Answer: Thank you for your detailed and good comments. Many studies have confirmed that in vitro allergenicity could be assayed by KU812 cells. We didn’t test to confirm that the morphology of basophils was exhibited by KU812 cells in the experiment. We will accept reviewers’ advice in the future study.

Q4.    To assess protein endocytosis the authors used mouse bone marrow-derived DCs, in this experiment do the authors include the step of washing cells upon incubation with fluorescently labeled proteins? Residual fluorescently labeled proteins could interfere with obtained results. Also Figure 5., in which these results are presented, I suggest being modified in a way that fluorescence intensity should be presented as bars and figure legend for different protein treatments should be added.

Answer:Thank you for your detailed comments. The pHrodo Green dye was used to label proteins as it fluoresces brightly at acidic pH with almost no fluorescence at neutral pH, which makes this dye a good indicator of the localization of protein or modified protein in the endolysosomal compartments. So, in this experiment, it is not necessary to wash cells, because residual fluorescently labeled proteins in outside the cells have no fluorescence. We modified the Fig 5. Fluorescence intensity was presented as bars and figure legend for different protein treatments.

Reviewer 2 Report

The authors subject conalbumin (CA) to oxidation and characterize the effects on CA structure and immunological properties.  Protein structure was assessed with circular dichroism (CD), intrinsic fluorescence (IF) and differential scanning calorimetry 56 (DSC).  Lipid peroxidation reduced the IgE-binding properties of CA complexes formed after treatment with 2,2′-azobis 38 (2-amidinopropane) dihydrochloride (AAPH) and acrolein.  While the study methods are well described and performed, there are several points that must be addressed. 

The text must be edited to improve reader comprehension and remove awkward wording and sentence structure, but there are too many instances to point them all out here    

The CD spectra in Figure 2A/B do not correlate well with the gel image in Figure 1.  For example, the 1mM AAPH and 0.1 mM acrolein treatments do not alter the migration/appearance of the CA on the gel, but yet they appear to alter the structure of the entire population of protein in the CD.  Can the authors explain the differences?  If the protein is so effectively modified by the treatments this should also be reflected in its migration on the gel even at low concentrations

The labels in Figure 2 may be mixed up with the test in the Figure 2 legend, and there are no letter labels on the data in Figure 2C 

The authors should include some pictures of the cells used in Figure 5, and the color scheme of the different samples is difficult to interpret.  The A25 sample does not appear to be consistent with the text on lines 252-252

Could the authors quantify the IgE binding of samples (including the acrolein treated CA IgE-binding) in Figure 6A and include the slower migrating forms?   How does this compare to the ELISA results in Figure 6B? 

Lines 340-342, what amino acids in CA were altered by the AAPH-CA and acrolein-CA and what were the resulting modifications? 

Author Response

Dear reviewer,

Thank you so much. We have carefully revised and checked our manuscript and answered all questions point by point. We highlighted the changes using the “Track Changes” function. We greatly appreciate your thoughtful advice and comments to help improve our study. We would like to re-submit the manuscript for consideration of publication. We would assure that all of the authors have read and approved the final submitted manuscript.

Q1. The text must be edited to improve reader comprehension and remove awkward wording and sentence structure, but there are too many instances to point them all out here.    

Answer:Thank you for your detailed comments. The editing of the English language had been modified carefully. We asked Dr. Ann Power Smith for editorial revision of this manuscript. Dr. Ann Power Smith is a professional English language editor for scientific and medical manuscripts and has 20 years' experience editing biomedical manuscripts, grants and other communications

Q2. The CD spectra in Figure 2A/B do not correlate well with the gel image in Figure 1.  For example, the 1mM AAPH and 0.1 mM acrolein treatments do not alter the migration/appearance of the CA on the gel, but yet they appear to alter the structure of the entire population of protein in the CD.  Can the authors explain the differences?  If the protein is so effectively modified by the treatments this should also be reflected in its migration on the gel even at low concentrations

Answer:Thank you for your detailed comments. CD spectra showed the secondary structure of the protein, comprised of four different conformations including α-helix, β-turn, β-sheet, and a random structure. 1mM AAPH and 0.1 mM acrolein treatments don’t alter the migration/appearance of the CA on the gel significantly. However, AAPH and acrolein could combine with protein, which could change the secondary structure, including α-helix, β-turn, β-sheet, and a random structure.

Q3. The labels in Figure 2 may be mixed up with the test in the Figure 2 legend, and there are no letter labels on the data in Figure 2C 

Answer:Thank you for your detailed comments. “Figure 2. Effect of AAPH-CA and acrolein-CA on (A) intrinsic fluorescence absorption, (B) CD spectroscopy” was changed to “Figure 2. Effect of AAPH-CA and acrolein-CA on (A) CD spectroscopy, (B) intrinsic fluorescence absorption”. Moreover, the letter labels on the data in Figure 2C were added.

Q4. The authors should include some pictures of the cells used in Figure 5, and the color scheme of the different samples is difficult to interpret.  The A25 sample does not appear to be consistent with the text on lines 252-252

Answer:Thank you for your good and detailed comments. We modified the Figure 5 to make it clear. The rate of endocytosis of modified protein was also increased with 25 mmol/L AAPH concentration. In this experiment, we only detected the fluorescence by a microplate reader without taking pictures of cells.

Q5. Could the authors quantify the IgE binding of samples (including the acrolein treated CA IgE-binding) in Figure 6A and include the slower migrating forms?   How does this compare to the ELISA results in Figure 6B? 

Answer:Thank you for your good comments. The authors were able to qualitative the IgE binding abilities in Figure 6A and semi-quantitative IgE binding abilities in Figure 6B. The bands of CA become shallow after AAPH and acrolein treatment in Figure 6A. In corresponding to the ELISA results, the OD value also decreased after AAPH and acrolein treatment in Figure 6B.

Q6. Lines 340-342, what amino acids in CA were altered by the AAPH-CA and acrolein-CA and what were the resulting modifications? 

Answer:Previous studies showed that the amino acid residues of the protein (Lys, Try, and His) are easily modified by lipid peroxidation (Refsgaard, H. H.; Tsai, L.; Stadtman, E. R. Modifications of proteins by polyunsaturated fatty acid peroxidation products. Proc Natl Acad Sci U S A, 2000, 97(2), 611-616. doi: 10.1073/pnas.97.2.611). Peptides 1–24, 40–60, 142–159, and 139–117 are major IgE binding epitopes of CA (Suprun, M.; Sicherer, S. H.; Wood, R. A.; Jones, S. M.; Leung, D. Y. M.; Burks, A. W.; Dunkin, D.; Witmer, M.; Grishina, G.; Getts, R.; Suárez-Fariñas, M.; Sampson, H. A. Mapping Sequential IgE-Binding Epitopes on Major and Minor Egg Allergens. Int Arch Allergy Immunol, 2022, 183, 249-261. doi: 10.1159/000519618), which contains many Lys, Try, and Leu residues. Therefore, lipid peroxidation led to the destruction of allergic epitopes, which could reduce the IgE-binding capacity of CA. We revised this section and added the new references in the revised manuscript.

Round 2

Reviewer 1 Report

The authors of the manuscript entitled “Functional and allergenic properties assessment of conalbumin (ovotransferrin) after oxidation” put effort into and significantly improved the manuscript text following a reviewer’s comments. An additional minor comment is referred to the subtitle 3.1., SDS-PAGE is not explanatory enough, suggesting that “CA oxidation products resolved by SDS PAGE” is a more suitable subtitle.